# Recent Advances in Surface Plasmon Resonance Imaging Sensors

**DOI:** 10.3390/s19061266

**Published:** 2019-03-13

**Authors:** Dongping Wang, Jacky Fong Chuen Loo, Jiajie Chen, Yeung Yam, Shih-Chi Chen, Hao He, Siu Kai Kong, Ho Pui Ho

**Affiliations:** 1Department of Mechanical and Automation Engineering, The Chinese University of Hong Kong, Hong Kong, China; lilianadpwang@gmail.com (D.W.); yyam@mae.cuhk.edu.hk (Y.Y.); scchen@mae.cuhk.edu.hk (S.-C.C.); 2Department of Biomedical Engineering, The Chinese University of Hong Kong, Hong Kong, China; jackyfcloo@cuhk.edu.hk (J.F.C.L.); super1cjj@gmail.com (J.C.); 3Biochemistry Programme, School of Life Sciences, The Chinese University of Hong Kong, Hong Kong, China; skkong@cuhk.edu.hk; 4School of Biomedical Engineering, Shanghai Jiao Tong University, Shanghai 200240, China; haohe@sjtu.edu.cn

**Keywords:** surface plasmon resonance, SPR imaging, optical design, biosensing, point-of-care sensor

## Abstract

The surface plasmon resonance (SPR) sensor is an important tool widely used for studying binding kinetics between biomolecular species. The SPR approach offers unique advantages in light of its real-time and label-free sensing capabilities. Until now, nearly all established SPR instrumentation schemes are based on single- or several-channel configurations. With the emergence of drug screening and investigation of biomolecular interactions on a massive scale these days for finding more effective treatments of diseases, there is a growing demand for the development of high-throughput 2-D SPR sensor arrays based on imaging. The so-called SPR imaging (SPRi) approach has been explored intensively in recent years. This review aims to provide an up-to-date and concise summary of recent advances in SPRi. The specific focuses are on practical instrumentation designs and their respective biosensing applications in relation to molecular sensing, healthcare testing, and environmental screening.

## 1. Introduction 

### 1.1. Brief History of Surface Plasmon Resonance

The phenomenon of anomalous diffraction due to the excitation of surface plasmon waves was first observed in 1902 by Wood when he shone polarized light onto a metal-backed diffraction grating [1]. In 1968, Otto [2] reported the attenuated total reflection (ATR) coupling method for surface plasmon excitation. In 1971, Kretschmann [3] presented the Kretschmann configuration of ATR coupling, which is the most widely used excitation method in current surface plasmon resonance (SPR) sensors. Surface plasmon microscopy was invented by Rothenhäuslar and Knoll in 1988 [4]. Plasmon surface polariton field was used to image microscopic interfacial structure. Phase interrogation SPR sensing was introduced in 1996 by Nelson et al. [5] and a steep slope of phase response over a narrow range of refractive index was observed. In 1998, Nikitin et al. [6] demonstrated the first phase-resolved SPR imaging (SPRi) sensor based on phase shift measurement in a Mach–Zehnder interferometer configuration. Following these pioneered works, research interests in SPRi have been growing rapidly and a variety of practical applications especially on biosensing started to emerge in the past two decades.

### 1.2. Operation Principle of SPR Biosensing

Surface plasmon resonance is a charge-density oscillation that may exist at the interface of two media with dielectric constants of opposite signs [7]. The simplest geometry in which a surface plasmon can exist consists of a semi-infinite metal with a permittivity, $\varepsilon_{m}=\epsilon_{m}^{\prime }+i\varepsilon_{m}^{\prime \prime }$, and a semi-infinite dielectric with a permittivity, $\varepsilon_{d}=\epsilon_{d}^{\prime }+i\varepsilon_{d}^{\prime \prime }$, where $\epsilon_{m}^{\prime }$ and $\epsilon_{d}^{\prime }$ are real parts of permittivity; $\epsilon_{m}^{\prime \prime }$ and $\epsilon_{d}^{\prime \prime }$ are imaginary parts of permittivity [8]. It is a transverse-magnetic (TM)-polarized wave which means its magnetic vector is perpendicular to the direction of propagation along the interface and parallel to the plane of interface. Surface plasmon wave is an evanescent wave with the intensity of magnetic field reaching a maximum at the interface and decaying into both the metal and dielectric media [9].

The propagation constant of a surface plasmon at the metal–dielectric interface can be expressed as
(1)$$k_{SP}=\frac{\omega }{c}\sqrt{\frac{\varepsilon_{d}\varepsilon_{m}}{\varepsilon_{d}+\varepsilon_{m}}}$$
where ω is the angular frequency, c is the speed of light in a vacuum [10]. Equation (1) represents a guided mode, surface plasmon, only if $\epsilon_{m}^{\prime }<-\epsilon_{d}^{\prime }$. Noble metals such as gold and silver exhibit a negative real part of the permittivity in the visible and near-infrared (NIR) wavelength range, thus are suitable for SPR sensing applications. The imaginary part of the propagation constant is associated with the attenuation of the surface plasmon along the direction of propagation. The established theoretical model suggests that the incident light wavelength determines the evanescent wave penetration depth [11]. The radiative losses of the plasmon wave decrease with longer wavelength resulting in a narrowing of the SPR response curve, and the propagation length of the surface plasmon wave is longer in the NIR range than that in the visible region. Therefore, the use of NIR light in SPRi provides an inherent higher sensitivity but lower lateral resolution than those achieved with visible wavelength in a prism-coupled SPR device [12]. An objective-based SPR microscope can be employed to recover the diffraction limit of SPRi in NIR wavelength range [13].

Light wave can excite surface plasmon only if its wavevector component parallel to the metal–dielectric interface, i.e., p-polarized light matches the propagation constant of the surface plasmon. The wavevector of p-polarized incident light can be described as
(2)$$k_{in}=\frac{\omega }{c}\sin\theta ,$$
where θ is the angle of incident light with respect to the metal–dielectric interface. Surface plasmon cannot be excited directly by light incident onto a smooth metal surface because the numerical value of $k_{in}$ is always smaller than that of $k_{SP}$. The wavevector of incident light can be increased in order to match that of the surface plasmon by the ATR via a coupling device, such as a glass prism. The wavevector is modified as
(3)$$k_{in}^{\prime }=\frac{\omega }{c}sin\theta \cdot \varepsilon_{p}^{1/2},$$
where $\varepsilon_{p}$ is the dielectric constant of the prism material. When SPR happens at a specific incident angle or wavelength, the energy of incident photons is coupled into surface plasmons, resulting in the attenuation of the total internal reflection light beam. It produces an absorption profile in the reflection spectrum, which is known as the SPR response curve. The position of reflection minimum, called SPR dip, implies the SPR resonance angle or wavelength, which is correlated to the refractive index (square root of the dielectric constant) in the dielectric medium. Refractive index changes in the vicinity of the interface can be detected by measuring changes in the SPR dip position, which is commonly used to characterize the SPR condition variations.

As described by the Fresnel model [5], the reflectivity of p-polarization and s-polarization are expressed as
(4)$$r_{p}=\left|r_{p}\right|e^{i\varphi_{p}}\;\;\mathrm{and}\;\;r_{s}=\left|r_{s}\right|e^{i\varphi_{s}},$$
where $\varphi_{p}$ and $\varphi_{s}$ are the phase of p-polarization and s-polarization, respectively. At SPR, $\varphi_{p}$ of the incident light is changed due to the energy transfer between light and surface plasmon with $\varphi_{s}$ unchanged. Hence, a phase difference between p-polarized and s-polarized light is generated [14], which forms the basic principle of phase- and polarization-based SPR sensors.

The phenomenon of surface plasmon can be utilized to make an SPR sensor. For SPR sensors, change in the refractive index of dielectric medium is the measurand, which changes the propagation constant of the surface plasmon. The change of propagation constant consequently alters the coupling condition between the light wave and the surface plasmon, which can be finally measured as a change in one of the characteristics of the optical wave interacting with the surface plasmon. SPR sensors can be categorized into intensity, angular, wavelength, phase, and polarization interrogated sensors based on which characteristic of the light wave is being measured [15].

Biomolecular binding events happening at the metal–dielectric interface forms an ultra-thin organic layer on the metal film, resulting in a change of refractive index in the vicinity of the interface, which builds up the mechanism of SPR biosensors [16]. In pursuit of high-throughput biosensing or screening capability, SPRi has been developed. Both SPR and SPRi sensors share similar detection principles, whereas SPRi embraces an additional merit of high throughput.

## 2. Optical Configurations of SPRi

To date, SPRi sensors based on intensity [4,9,17,18,19,20,21,22,23,24,25,26,27,28,29,30,31], angle [32,33,34,35,36,37,38], wavelength [39,40,41,42,43,44,45], phase [6,46,47,48,49,50,51,52,53,54,55], and polarization [28,38,56,57,58,59,60,61,62,63] interrogation have been widely reported.

### 2.1. Reflectivity-Based SPRi

Reflectivity-based SPRi is a straightforward format for two-dimensional (2D) array sensing. It was first invented by Rothenhäuslar and Knoll [4] in 1988 and henceforth intensive research work on reflectivity-based SPRi have been conducted [19,20,21,22,23,24,25,26,27,28,29,30,31]. It is able to perform parallel measurements in up to hundreds or even thousands of sensing spots [17,18] with a typical resolution of 10^−5^ refractive index units (RIU) [9,26]. The operation principle is explained in Figure 1. Monochromatic light is used as the excitation source. The optimum angle of incident light is chosen so that the system is operating in the “linear” response region of the SPR angular response curve, as defined in Figure 1. Shift of SPR dip is quantitatively proportional to the change of refractive index at the metal dielectric interface, which is then translated into the change of reflectivity in the linear region of the SPR angular response curve [64]. Reflectivity-based SPRi therefore creates a 2D intensity contrast image of the refractive index distribution of the sensing surface [26].

In 1997, Jordan et al. [19,20,21] used monochromatic He-Ne laser as the SPR excitation light source where the expanded beam impinged on the gold sensing surface at the resonance angle and SPR image was subsequently captured by a monochromatic charge-coupled device (CCD) camera. Later on, Nelson et al. [22] improved the performance by using a NIR light source. Sharper SPR angular response curve was obtained with the NIR light compared with that of visible light, resulting in a higher-contrast SPR image. Utilization of an incoherent white light source and a narrow band-pass filter allows for the easy adjustment of excitation wavelength and eliminates the laser fringes that have been observed in conventional SPRi set-ups. Using incoherent light excitation, Wark et al. [23] presented a long-range surface plasmon (LRSP) imaging system which was created on a thin gold film embedded between two identical dielectrics. LRSP possesses longer surface propagation length, higher electric field strength, and sharper angular SPR curve than conventional SPR. Fu et al. [24,25] reported a reflectivity-based SPRi system which used a tilted interference filter to choose the optimal excitation wavelength for a given sample. The resolution was found to be 3 × 10^−5^ RIU. 

So far, reflectivity-based SPRi has been widely used for high-throughput analysis in biosensing applications [27,28,29,30]. Nevertheless, reflectivity-based SPRi has become a high-throughput technique through compromising detection sensitivity due to the adoption of a less compact optical configuration, less sensitive mode of a single point intensity-based measurement, and commonly less-sensitive detectors [65]. Furthermore, for the detection of multiple interactions simultaneously, a homogeneous and optimal response for all sensing spots cannot be obtained with the choice of a unique operating point [31].

### 2.2. Angle-Resolved SPRi

In angle-resolved SPRi, the incident angle is continuously scanned with wavelength fixed. Shift of SPR angular dip is then measured for different refractive index samples or molecular binding events. The SPR dip shift is the only reliable and underived parameter that directly reflects the corresponding mass change on the SPR sensing surface [34,66]. Angle-resolved SPRi normally possesses a higher resolution and a wider dynamic range compared with reflectivity measurement-based SPRi. 

One-dimensional (1D) angle-resolved SPRi was developed in 2001 [33] with one-dimension measuring SPR reflectivity versus angle and the other dimension producing a 1D image of the sensing surface. In 2005, Wolf et al. [32] developed a prism and detection arm rotated 2D angle-resolved SPRi system for quantitative characterization of biological response with a limit of detection (LOD) of refractive index change of 0.002. In 2007, Ruemmele et al. [34,35] reported an inexpensive automated angle-scanning SPRi instrument. The schematic of this angle-resolved SPRi is illustrated in Figure 2. They achieved a linear relationship between the SPR dip shift and the mass density change on the sensing surface over a wide angle range of 8° [35]. Zhou et al. [37] developed another angle-scanning SPRi device using a rhombic structure to convert the linear motion of a piezoceramic motor into the angular motion of laser and CCD arrays. Thanks to the fast scanning speed provided by the piezoceramic motor, authors could detect the mismatched bases in the caspase-3 DNA at a rapid and high-throughput fashion. In 2011, Liu et al. [38] developed a 2D polarization interferometry based on parallel line scan angle interrogated SPRi technique. They combined the angle interrogation based SPR sensing and the parallel line scanning into a 2D quantitative SPR sensor. Furthermore, polarization interferometry technique was applied to lower the minimum of the SPR dip and reduce the noises related to the light intensity. Resolutions of 1.4 × 10^−6^ RIU under normal condition and 4.6 × 10^−7^ RIU under a more time-consuming condition were achieved.

### 2.3. Spectral SPRi

In the wavelength interrogation mode, the incident angle is fixed while the SPR spectral profile and the SPR spectral dip are obtained by either continuously scanning the incidence wavelength or using a spectrometer. Spectral SPRi provides a wide dynamic range, and the resolution could be as high as that offered by the angle interrogation mode. 

In recent years, a lot of efforts have been made to improve the throughput in wavelength interrogated SPR. In 2006, Fu et al. [39] reported a 1D SPRi system based on the wavelength interrogation. Light with wavelength distribution along one spatial dimension was generated by a linear variable filter. Resolution of 7.5 × 10^−4^ RIU was obtained which is limited by the bandpass function of the linear filter. In another 1D SPRi spectroscopy instrument, Lee et al. [45] reported a full spectral image from a single nanohole array chip with 50 parallel microfluidic channels. A slit and a grating were used to disperse spectrum in one dimension and the other dimension corresponds to the spatial distribution direction of multi-channels, as shown in Figure 3. A CCD camera was used to record the reflected light from the grating in real time. The resolution of the proposed SPRi system is 7.7 × 10^−6^ RIU. Yuk et al. [40] reported a 2D spectral SPRi sensor in combination with the position control of the optical fiber probe of a spectrometer. The scanning time for a spot with 2 mm diameter is 180 s, and the sensor resolution is 7.6 × 10^−5^ RIU. In addition to high throughput, it is also important to achieve real-time kinetics monitoring of biological interactions. To improve the time resolution, Liu et al. [41] reported an optical line scan spectral SPRi system. A 2D refractive index distribution of the entire sensing surface could be obtained with a 1D optical line parallel scan. It took 60 s to image a 2D array with an area of 8 × 8 mm^2^. To further speed up measurements, an optimized algorithm based on five parameters was developed for spectral SPRi sensors. It took 10 s to measure the whole 2D arrays [42]. In 2016, Zeng et al. [43,44] built a fast-wavelength interrogation SPRi system with a liquid crystal tunable filter for incident wavelength scanning. A high scanning speed was achieved by dynamically adjusting the scanning wavelength range to match the SPR dip using a feedback loop. With this technique, about 0.7 s was required for one cycle of scanning. They achieved a resolution of 4.69 × 10^−6^ RIU and a dynamic range of 5.55 × 10^−2^ RIU [44]. In 2017, the same team reported a high-speed spectral SPRi biosensor using an acousto-optic tunable filter (AOTF) and a white light laser. An SPR dip measurement was completed within 0.35 s. A dynamic range of 4.63 × 10^−2^ RIU and a resolution of 1.27 × 10^−6^ RIU were achieved [67]. In 2018, Bak et al. [68] demonstrated a real-time spectral SPRi based on a wavelength-swept laser. The proposed system has a higher output power for a large-area illumination (12 × 12 mm^2^) compared with that offered by the white light source-based SPRi system. A dynamic range of 7.67 × 10^−3^ RIU and a resolution of 1.89 × 10^−6^ RIU were obtained with a scan rate of 12 Hz in the proposed system.

### 2.4. Phase-Resolved SPRi

The phase interrogation SPR technique requires a fixed angle and wavelength of incident light while the phase difference between the signal beam and reference beam is measured. Compared with angle or wavelength interrogation mode, the phase interrogation mode offers ultrahigh sensitivity. Meanwhile, it also holds great potential for integration with imaging technology to achieve a high throughput [69]. Interference between a signal beam and a reference beam is normally obtained in phase interrogated SPR to calculate the phase shift information.

In 1996, Nelson et al. [5] introduced the phase detection for SPR sensing which provides three times higher resolution compared to angular and spectral SPR sensors. In 1998, Kabashin et al. [6] demonstrated the first phase-resolved SPRi sensor. The probe and reference beam interfered in a Mach-Zehnder interferometer configuration, and the phase information over the sensing surface was captured by a CCD camera. The sensor resolution was found to be 4 × 10^−8^ RIU for gas detection. Later in 2000, the same group [46] introduced an interferometric SPRi sensor for micro-array bio and chemistry sensing. To enhance long-term stability and suppress external disturbances, Su et al. [47,48] demonstrated a common-path phase shift interferometry-based SPRi sensor. A liquid crystal phase retarder was used for phase modulation, and the phase information was decoded with a five-step phase shift reconstruction algorithm. The sensor resolution was found to be 2 × 10^−7^ RIU in nitrogen and argon gas detection. Xinglong et al. [49] demonstrated a common-path interferometry-based SPRi sensor by combining SPR and spatial phase modulation measurement. The SPRi sensor in an array format was then demonstrated for antigen–antibody binding process with a resolution of around 10^−6^ RIU [50,51]. It has been explored to broaden the detection range of phase interrogated SPRi [52,53,70]. Wang et al. [52] developed an SPRi biosensor based on differential interferometry for protein microarray experiments with a resolution of 5 × 10^−7^ RIU over a wide dynamic range of 0.015 RIU. Shao et al. [53] proposed a wavelength-multiplexing phase-sensitive SPRi sensor. As shown in Figure 4, a liquid crystal tunable filter was used to sweep the input wavelength from a white light source and a liquid crystal modulator to modulate the phase retardation. With this system, they were able to measure SPR phase at any wavelength of interest, achieving a wide dynamic range of 0.0138 RIU with a resolution of 2.7 × 10^−7^ RIU. Wong et al. [54,55] presented a 2D SPR phase imaging technique where a piezoelectric transducer was used to modulate the phase in the time domain. A 2D SPR phase map was obtained with every pixel on the SPR image corresponding to the phase shift value. They separated the light interference pattern into the p-polarized and s-polarized images which eliminated the common optical path noise and a resolution of 8.8 × 10^−7^ RIU has been demonstrated.

### 2.5. Polarization Contrast-Based SPRi

In the polarization contrast-based method, linearly polarized incident light was used to excite SPR, which results in both the amplitude and phase of p-polarized reflected light altered whereas s-polarized light not affected [15]. Thus, the polarization of the reflected light is no longer linear but elliptical. Elliptical polarization corresponding to a certain refractive index at the sensing surface is made extinguished using a λ/4 waveplate and a polarizer. When refractive index changes, the polarization of the reflected light alters and the transmitted light intensity through the λ/4 waveplate and the polarizer increases. Measurement of the intensity variation is the operation principle of the polarization contrast-based SPR sensors [71]. 

Kabashin et al. [72] pioneered the phase and polarization transformations that occurred in SPR and proposed the usage of the SPR-related phase jump as a resonance point marker to improve the signal pattern contrast. Later on, Piliarik et al. [28,56,57] reported a polarization contrast SPRi sensor with the subtraction of the dark current signal and intensity fluctuations of the light source with 108 sensing channels. They demonstrated a resolution of 2 × 10^−7^ RIU [57]. Patskovsky et al. [58] described a scheme of spatially modulated SPR polarimetry. A birefringent wedge was utilized to produce periodic changes of phase relations between the p-polarized and s-polarized light and the phase-polarization information was then extracted by the Fourier transform method. This method enables the combination of ultra-high phase sensitivity with good signal-to-noise (S/N) background. A resolution of 10^−6^ RIU was obtained which is limited by external temperature and intensity drifts. Since 2011, Wong et al. [61,62,63] have developed spectral-phase SPRi sensors based on polarization control scheme. As shown in Figure 5, two polarizers with perpendicular transmission axes are configured in series so that the transmission of an incident beam is forbidden. At the excitation wavelength of surface plasmon, a phase difference is introduced between p-polarized and s-polarized light, which rotates the orientation angle of the polarization ellipse, allowing the light interacting with the surface plasmon to pass through the crossed polarizers. Since only a particular region of wavelength can excite SPR, a particular spectral profile is generated and the corresponding color change is captured in the spectral SPR image. A sensor resolution of 2.7 × 10^−6^ RIU was demonstrated.

## 3. Recent Developments in SPRi Instrumentation

### 3.1. Solid-State Angle-Resolved SPRi

In order to avoid any mechanical scan, which may have unavoidable errors such as backlashes, solid-state angle-scanning strategies for angle-resolved SPRi have been developed, which enables a high-speed, agile, robust and accurate angle-scanning with no moving parts. In 2007, VanWiggeren et al. [60] developed an angle-resolved SPRi sensor using an acousto-optic deflector (AOD). In this sensor, collimated laser light fills the aperture of the AOD and then deflected by the AOD. Deflected beam illuminates the sensor surface with varying incident angle scanned by the AOD. Light reflected from the sensor surface is imaged onto a CMOS camera. This method ensures the illuminated region on the sensing surface of the prism remains unchanged, and the image of the sensing surface does not move across the detector array during the angle scanning by the AOD. With this sensor, interaction kinetics were sampled at a 10 Hz rate, and the dynamic range of 1.32 to 1.38 RIU was obtained.

In 2014, Zhang et al. [73] introduced a real-time non-scan multichannel angle-resolved SPRi method. The light beams pass through a cylindrical lens array (CLA) and are focused to three narrow lines on the sensing surface which correspond to three imaging channels. An image captured with a CCD camera probes SPR angular response from the three channels. This SPRi configuration has a resolution of 2.7 × 10^−5^ RIU. The symmetrical optical waveguide (SOW) sensing structure with sharp resonance curve was applied to improve the system resolution. The imaging system achieved a resolution of 9.4 × 10^−6^ RIU with the SOW sensing structure.

In 2018, Wang et al. [74] reported a digital micro-mirror device (DMD)-enabled angular interrogated SPRi sensor, as shown in Figure 6. DMD is a high-speed 2D light modulation device containing several millions of micromirrors. Angular scanning is achieved by selectively switching the columns of micromirrors on the DMD to the “on” or “off” positions. In this respect, angular interrogation of SPR is realized in the time domain. The reflected light from the sensing surface is collected by a single-point photodetector. The unique capability of DMD enables a high-speed, scanless, and programmable SPR sensor design that facilitates a single-point photodetector for measurement with a high S/N ratio, wide dynamic range, and fast response, thereby achieving high-resolution SPR measurements. By dividing the DMD into multiple regions along the horizontal direction, four-channel SPRi has been demonstrated. The experimental results have verified a resolution of 3.54 × 10^−6^ RIU and a detection limit of 9 ng/mL.

### 3.2. Microscope Objective-Based SPRi

In order to achieve diffraction-limited spatial resolution and distortion-free angle-resolved SPRi, a high numerical aperture microscope objective has been used for the excitation of SPR [75,76,77,78]. First proposed by Huang et al. [75] in 2007, the microscope objective-based SPRi system was built on the basis of the Kretschmann configuration using a high numerical aperture oil immersion objective and an inverted microscope. As shown in Figure 7, the laser from the fiber output is collimated and focused onto the back focal plane of a high numerical aperture objective, and finally impinges onto the sensing surface as a parallel beam out of the objective. A linear translation stage is used to adjust the offset of the laser from the optical axis of the objective, thus the incident angle with respect to the sensing surface for SPR excitation is scanned. Reflected light is collected by the objective and then imaged onto a CCD camera. The employment of a high numerical aperture and high magnification imaging system ensures that the resolution of the imaging optics is diffraction-limited (~300 nm) [75].

Following this, a holographic SPRi method based on microscope objective for simultaneous amplitude and phase-contrast SPRi has been constructed [76]. In the optical configuration, an inverted SPR microscope with a Mach-Zehnder interferometer is placed in the detection arm. The s-polarized beam passes through a λ/2 plate and interferences with the p-polarized beam. The interference pattern of the two beams forms the holographic image which is recorded with a CCD camera and further analyzed by the numerical processing. The incident angle onto the sensing surface is changed using a linear translation stage. This method improved the dynamic range and throughput with respect to both phase-contrast and amplitude-contrast imaging methods.

In another microscope objective-based SPRi module, a digital light projector as the incoherent excitation light source has been adopted for the selection of wavelength or angle of the incident light [79,80]. This SPR microscope provides the flexibility to either choose measurements of the sample on the back focal plane of the objective, which provides angle-dependent reflectivity data; or on the imaging plane of the sensing surface, which provides the image data at a single optimized angle of incidence. 

### 3.3. Nanoparticle/Nanostructure-Based SPRi

With the development of nanotechnology, nanoparticles (NPs) and nanostructures have been intensively incorporated in SPR applications to enhance the performance of SPR techniques. Among these strategies, metallic NPs have been most commonly used for signal amplification by two-fold contributions: coupling effect and mass increase as amplification tags [81,82,83,84]. The first contribution comes from the coupling effect between the surface plasmon polaritons from the conventional SPR sensor chip and the localized surface plasmon from NPs. Typically, for a metallic NP with a size of several tens of nanometers or smaller, the free electrons are trapped locally on the NP surface. Under proper optical excitation, these free electrons would oscillate collectively and thus results in localized SPR (LSPR) which is sensitive to the localized dielectric environment, size, shape, and composition of the metallic NPs [31]. Signal enhancement from amplification tags lies in the fact that, upon addition of linked NPs, a pseudo mass increase of analyte is induced. This mass change finally produces a higher refractive index change on the SPR sensing surface. Different metallic NPs including Au NPs [82,85,86,87], Ag NPs [81], Pd NPs [88], and Pt NPs [89] have been applied to increase SPR sensitivity in various biomolecular detection applications. In addition to metallic NPs, other particles, for instance, SiO_2_ NPs [90,91], magnetic NPs [92,93], quantum dots (QDs) [94,95,96], and graphene [97,98,99,100,101] have also been used in SPR sensitivity enhancement and signal amplification for their distinct superior properties. Besides NPs, SPRi platform based on various nanostructures has been extensively investigated as well, such as nanohole array [102,103], nanoslit array [104,105,106], nanodonuts array [107], nanorings array [108], randomly distributed nanostructures [109,110,111,112], and 3-dimensional (3D) plasmonic metamaterial [113] for high-throughput and high-sensitivity SPR sensing applications. Nanostructures-based SPRi sensors are inherently miniaturized versions of SPR sensors with merely a simple measurement of transmittance, eliminating the need of either a prism or complex optical system. Besides, it offers unprecedented capabilities for multiplexed assays as the sensing area is limited by the size of the nanostructure, which virtually expands the throughput to the nanostructure level [114]. The parallel SPRi system using nanostructure arrays is easily integrated with microfluidics for automatic delivery of multiple sample flows, providing the precise control over each reaction stage and increasing the throughput of a single chip [115]. The integration of NPs and nanostructures with microfluidics in SPR sensing could bring lots of promising opportunities for future developments in biosensing applications [31]. 

### 3.4. Smartphone-Based SPRi

In 2008, Schasfoort and Schuck took on the challenge for SPR to be taken as a technology suitable for point-of-care (POC) diagnostics [116]. Since then, efforts have been made to build more portable, affordable and accessible SPR sensor products for POC applications. Taking advantage of the wide accessibility of smartphone, smartphone-based SPR platforms have been developed for portable, on-site and remote healthcare testing in the POC and household devices for daily biosensor applications in resource-bound conditions [117]. In 2012, Preechaburana et al. [118] demonstrated an angle-resolved SPR detection system based on a single disposable device which was configured to use conditioned illumination and optical detection from a smartphone. A resolution of 2.14 × 10^−6^ RIU has been achieved. Later on, fibre-optic SPR biosensors based on smartphone [119,120] have been reported. All optical components and the sensing element are connected by optical fibers and fixed on a smartphone case. The LED flash light acts as the light source and the SPR image is recorded using the smartphone camera. Smartphone-based fibre-optic SPR sensors make POC sensors suitable for practical uses, thanks to the advantages over conventional SPRi system, including low cost, light weight, compact size and simple installation capability, despite its high resolution of 7.4 × 10^−5^ RIU. LSPR sensors based on the smartphone platform have been developed in the past few years. The application of gold NPs, gold nanorods [121,122] and nano Lycurgus cup array [123] have been explored to provide a portable inexpensive platform with improved sensitivity and LOD. For high-throughput and multiplexed detection, Guner et al. [124] developed a smartphone-based SPRi platform for on-site parallel biosensing, which is illustrated in Figure 8. An SPRi platform with 4 × 4 spots has been demonstrated with a resolution of 4.12 × 10^−5^ RIU. For miniaturized smartphone-based SPR devices in which the entire process from sampling to sensing must be integrated on a chip, the incorporation of advanced microfluidic techniques may equip SPR sensors with the capability to work as an all-in-one POC diagnostic device [115].

## 4. Biosensing Applications

Biosensing is one of the important applications relying on the sensitive sensor nowadays. Owing to its advantages, e.g., highly sensitive, real-time and label-free, biosensing applications range from basic scientific research to the development of POC sensors. Thanks to the extensive investigations on SPRi, multiplex biomarker screening in a high-throughput manner could be achieved. For example, the automatic spot identification method using the combination of video accessing, image enhancement, image processing and parallel processing approaches to SPRi can accurately and quickly track multiple spots of interest in an SPR sensor array, regardless of the spot pattern, background contrast, light non-uniformity and defects [125]. In this session, we summarize the up-to-date practical approaches to biosensing applications, including molecular sensing, healthcare testing and environmental screening with the well-established conventional SPRi sensors, as well as novel SPRi setups.

### 4.1. Molecular Sensing

Molecular sensing involves the sensitive sensing of target molecules with low concentration. Although chemical and biochemical methods are available, they require multiple steps, such as reagent mixing, enzymatic amplification in order to obtain high sensitivity. SPRi is therefore an attractive alternative for conventional assays. SPRi enables a simultaneous label-free study of multiple biomolecular interactions, such as the binding affinity and kinetics, between the loaded analytes and the coated targets. Prior to the availability of the SPR technique, molecular biologists exploited traditional immunoassays, such as enzyme-linked immunosorbent assay (ELISA), to investigate the molecular interaction between different biomolecules. However, these methods required time-consuming process, such as fluorescence and radioactivity labeling, preparation of labeled molecules, multiple steps of reactions and post-data analysis. In addition, the sensitivity is affected by the auto-fluorescence of the analytes and the incubation time of the assay, while the specificity is contaminated by the non-specific interaction between the label. On the contrary, the high sensitivity and specificity with a high percentage of correlations between results obtained in the conventional assay ELISA and the developed SPRi of the label-free SPRi method is supported by an established protocol, and it is therefore the solution for molecular sensing work [126].

For ultrasensitive molecular sensing, the 2D spectral SPRi sensor using phase interrogation has been shown to offer better performance, with a detection limit down to 8.26 ng/mL (125 pM) in sensing the interaction between the anti-bovine serum albumin (anti-BSA) and the BSA antigen [53,61,62]. Moreover, the measurement time has been drastically shortened, such as using the liquid crystal tunable filter and the step size optimization that increase the interrogation speed and decrease the noise simultaneously, to below 1 s for multiplexed high-throughput data collection from real-time 2D SPR images, with the demonstration on real-time multiple detection of goat anti-rabbit IgG and rabbit IgG antigen–antibody interaction [43,67]. These proof-of-concept novel SPRi schemes for rapid and sensitive molecular sensing based on protein-protein interactions support their application in drug screening and biomarker detection. For example, the leakage of cytochrome c, a cell death marker from apoptosis, from cancer cells can be detected with a high sensitivity in spectral SPR and conventional SPRi [127,128]. Apart from antigen detection, antibody secretion from hybridoma cells can be quantified by using SPRi [129].

Another area in molecular sensing is nucleic acid analysis, which relies on the interactions by specific base-pairing between the DNA or RNA sequence of the surface-coated probe and the target. A spatially-resolved SPRi has been developed for the sensitive detection of short oligonucleotides, with a detection limit of 100 pM, which is a 100-fold improvement compared to conventional SPRi [130,131]. An important outcome of the basic research is the discovery of new recognition elements, such as new aptamer selection via SPRi technique. Aptamer, a single strand DNA or RNA with an intra-strand self-folding for a unique 3D configuration that specifically binds to the target, has shown its advantages including low cost and high stability, thus providing an alternative to conventional antibodies. The aptamer is selected from a DNA or RNA library of random long-length nucleotides such as 40 bp, resulting in a high variation of the nucleic acid sequence. The library is subjected to the interaction with target molecules in various conditions, such as an increase in salinity and a decrease in the library concentration with cycles, which is named as systematic evolution of ligands by exponential enrichment (SELEX), as indicated in Figure 9A–C. This results in an aptamer bound with the target analyte with the highest sensitivity and specificity. With the help of SPRi, the real-time kinetic profile of potential aptamer bound to the target can be revealed. It is found to be as low as six SELEX cycles with additional negative selection enough to obtain an aptamer highly bound to the target [132]. Yet, the recognition element-target interaction depends on environmental conditions, such as salt content and pH value. Therefore, buffer conditions screening for optimizing wash and elution steps in an affinity-based purification process, such as affinity chromatography in the early stage of developing new recognition elements, or even therapeutic recombinant ligands, such as recombinant insulin, production and purification has also been demonstrated with SPRi [133].

In addition to the discovery of new recognition elements, the investigation on the antigenic structure of herpes simplex virus for simultaneously measuring a larger number of protein-protein interactions in cross-competition or “epitope binning” studies provide new insights into the antigenic structure for understanding how vaccination with a particular subunit results in an immunity to the infection, and leads to a better vaccine development [135]. Besides studying the molecular interaction, high-throughput kinetic profiling, such as for epigenetic interactions, i.e., chemical modifications on histones and DNA/RNA, which is essential in epigenetic regulation on multiple body functions, could be investigated with SPRi as well [136].

### 4.2. Live Cell Analysis

Studying cell-surface interactions is another major biological research area. Conventionally, the investigation of cell-surface interactions usually involves fluorescence-labeled cells with observation via microscopy. The detection sensitivity and the reliability are limited by the capacity, lifetime and toxicity of florescence staining. Alternatively, SPRi on cell-surface interactions on the sensing surface supports high-resolution imaging and sensing. In the live cell imaging, the cell is first attached on the SPRi sensing surface, such as a gold surface. Using random nanodot arrays has enhanced the image resolution [110,137,138]. Time-lapse scanning surface plasmon microscopy of adherent living cells, bacterial biofilms have been demonstrated [139,140,141]. After the attachment, different assays have been demonstrated. Quantification of cell-to-substrate separation in human aortic endothelial cell (HAEC) culture in SPRi achieved a sensitive determination of the separation distance at 40–60 nm [142]. Simultaneous observation and quantification of protein layers, such as time-dependent deposition of the cellular protein, the change in drug-induced protein layers, as well as detailed cellular features, is used to understand the interactions of cells with the extracellular matrix (ECM) environment [143]. The detection sensitivity of the deposited protein is 20 ng/cm^2^, with a lateral resolution of 2 μm. In Figure 10, the deposition of fibronectin (FN), a glycoprotein that is an essential component of the extracellular matrix (ECM), is imaged and quantified. FN present in the plasma acts as a potential biomarker indicating cell adhesion, growth, migration, differentiation, wound healing and embryonic development, and the quantification of the fibronectin concentration shows a dynamic response range between 5 and 400 ng/mL, with a detection limit of 1.5 ng/mL that supports a reliable screening of patients (601 ± 72 g/mL) from healthy donors (140 ± 25 g/mL) [126]. 

Similar to molecular sensing, SPRi sensing surface pre-coated with antibodies can be used for studying cell surface antigens, thus constituting a useful tool for cell identification and blood group typing [144]. A T/S detection strategy using cell sedimentation (S) followed by a specific upward response (T) was employed to detect the specific binding of cells to the sensing surface [145]. Notably, real-time monitoring of cellular pathways inside a living cell is an interesting area. For example, monitoring G protein-coupled receptor signaling and its modulation to detect PKC translocation initiated by the ligand binding to mGluR1 as well as A1R-mediated modulation of mGluR1-mediated PKC translocation in cultured kidney-derived HEK293 cell line cells on the SPRi sensing surface was used to study the cellular basis for cerebellar motor learning [146]. Also, secretory defects in vascularized micro-organs, known as the islets of Langerhans, could result in diabetes. Studying the paracrine interaction in islets’ cells, an ideal candidate for detecting multiple islet’s secretion products using an SPR hormone array, is a promising way to understand the mechanism of secretory defects [147].

### 4.3. Healthcare Testing

Healthcare testing is another important application in biosensing. Nucleic acid and protein markers are usually the target biomarkers for rapid disease screening. For nucleic acid biosensing, hybridization to capture targets of DNA and RNA onto a DNA probe in SPR array is a common method [148]. Sensing of small nucleic acids, such as human microRNAs (hsa-miR), is challenging, since their sizes, i.e., mass, are much smaller than long chain DNAs, and their concentrations in biological fluids, such as saliva and the serum, are in femto- or pico-molar level. Traditional reverse transcription polymerase chain reaction (RTPCR) is used with additional pre-amplification steps, resulting in a 2-h assay, which is time-consuming for POC use. Therefore, SPRi on microRNA sensing has been investigated in order to provide a fast and efficient sensing method. RNase H, which digests the RNA probe of the recognition element in the bound DNA-RNA strand only and releases the DNA analyst to bind with another RNA probe, is an innovative method to amplify SPRi signals when reverse transcribed RNA is recognized and captured by the specific RNA probe, as shown in Figure 11A,B [149]. 

The detection of protein biomarker detection usually relies on antigen–antibody interactions on the SPRi sensing surface. For example, allergy screening with IgE on peanut sensitization confirmation based on levels of Immunoglobulin E (IgE) antibodies against four peanut epitopes and anti-IgE in SPRi has been demonstrated, as illustrated in Figure 11C,D [150]. The antigen–antibody interaction was extended to detect viral surface protein. Group-specific antibodies have been reported to identify highly divergent virus strains of rapidly evolving viruses in SPR nano-arrays [151]. Virus screening is usually performed using nucleic acid sensing, but the issues in rapid screening of the strains are demonstrated in the above SPRi. 

For cancer diagnosis, serum collagen type IV (COLIV), a promising tumor marker overexpressed in the serum of patients with colorectal, gastric, lung, liver and breast cancers, was detected with anti-human collagen type IV antibodies in SPRi, with a high specificity against potential interference substances, including albumin, collagen IV, laminin-5, heparin and glycoprotein GPIIB/IIIA in human samples [126,152]. The toleration in the presence of 100-fold excess of the potential interference substances supported its translational use in clinical cancer diagnosis.

Transfusion of incompatible red blood cells (RBCs) could result in adverse reactions in the presence of antibodies, as well as alloimmunization that complicates future transfusions. Therefore, prior to the transfusion of the donor blood of a matching blood type, the blood type of the recipient is determined with the agglutination-based RBC-typing method, a current gold standard for testing a selected number of antigens for ABO and Rh blood type determination. However, the drawbacks of time-consuming on agglutination reaction and the incapability of high-throughput identification of multiple blood group antigens hinder the up-to-date multiplex blood typing. Generation of an SPRi array with clinically relevant blood group antibodies, i.e., anti-A, -B, D, C, c, E, e, and K blood group antigen-specific antibodies, has been validated for screening A, B, and Rh blood groups in 946 donor samples with 100% agreement in cross-match comparison with Sanquin National Screening laboratory typed with classical serology (Figure 11E,F) [154]. Furthermore, it is found that, although the Rh antigen possesses a stronger interaction than A, B and AB antigens, it requires a longer incubation time to interact with the immobilized antibody than A and B antigens do due to the difference in the antigen type and their locations on the RBC [153].

Meanwhile, healthcare testing preferably performs in a POC sensor that is portable for on-site screening. Mobile-based SPRis have been developed thanks to the prevalence of smartphones with a rapid growth in the computing power which is capable of capturing a large amount of SPRi information with storage and analysis, either from the built-in processing power and memory or the cloud computing for high throughput on-site disease screening. On the other hand, coupling SPRi with other instruments such as mass spectrometry could help in both sensitive detection and accurate characterization of the target such as insulin degrading enzymes [155].

### 4.4. Environmental Screening

Environmental sensing is another major application of biosensing, especially for screening contamination from biological toxins and pathogens. Toxin and pathogen contamination in food are very common. For example, mycotoxin, such as deoxynivalenol (DON) and ochratoxin A (OTA), is one of the common toxins found in food and beverage, such as beer [156]. The “masked” mycotoxins that reveal their toxicities after conjugation with sugar or organic acid is worth for detection [157]. Portable SPRi has been used to measure these mycotoxins, with LOD of 17 ng/mL for DON and 7 ng/mL for OTA, representing less than 10% depletion of a tolerable daily intake of DON and OTA by beer. On the other hand, the detection of T-2 toxins and T-2 toxin-3-glucoside (T2-G) in wheat, with an additional amplification strategy using conjugated secondary antibody-gold NPs, yield the LODs of 1.2 ng/mL of T-2 toxins and 0.9 ng/mL of T2-G, which enables a sensitive detection of the toxin levels in wheat, at as low as 48 and 36 μg/kg. Besides, the concern over NP contamination in the environment is increasing. Rapid quantification and characterization of NP size at sub-ppb level (100 pg/mL) have been demonstrated in complex products such as wines, fruit juices, or cosmetic formulation with SPRi [158]. For screening bacterial contamination, salmonella detection from chicken carcass rinse samples and indigenous microflora has shown a detection limit of 6.8 CFU/mL in detecting 5 out of 6 Salmonella outbreak serotypes, with a good specificity against 6 non-Salmonella species (Figure 12A,B) [159]. 

Gas sensing is a recent popular trend. Thanks to the high S/N ratio of SPRi and the robustness of performance without a pre-equilibrated sensor, such as a high temperature pre-heating of semiconductor in conventional electronic gas sensors, SPRi has been explored extensively for “electronic nose” application. Atmospheric environmental monitoring of various harmful gaseous chemical mixtures, such as benzene, toluene, ethylbenzene, and xylene (BTEX) mixture, are important due to their wide uses in chemical plants [161]. Monitoring volatile organic compounds (VOCs), such as alcohol (1–6) and carboxylic acid family (I–V) with varying carbon chain lengths by SPRi provide comprehensive information on the composition of VOCs besides simple detection [162]. A good sensing selectivity, which is not limited to pure VOCs, is achieved by using PCA and HCPC as the efficient method of data analysis on multiple analysis of VOCs mixtures, as shown in Figure 13. Furthermore, the integration of SPRi to the micro-gas chromatography system allows a simultaneous separation and multidimensional detection of the target chemical in a gas mixture, leading to an increase in sensitivity or process to downstream procedure such as chemical selection, extraction and purification from a gas mixture [143,161].

## 5. Conclusions and Future Perspectives

We have reviewed the conventional SPRi configurations and the emergence of new SPRi designs developed in recent years. SPRi has found a variety of applications for its label-free, real-time, high-sensitivity and high-throughput characteristics. The applications on molecular sensing, healthcare testing and environmental screening have been highlighted. This promising technology is continuously evolving and innovating to improve the overall performance of SPRi. The sensitivity is a fundamental feature of the SPRi biosensor. From the perspective of noise properties of SPR system, because of its higher S/N ratio, wide dynamic range and fast response, a single-point photodetector offers higher sensitivity than a CCD camera. Moreover, solid-state design eliminates noises introduced by mechanical scan, providing a fast, robust and accurate SPR sensor module. From the perspective of signal amplification, significant growth in the area of using nanomaterials and nanostructures has further enhanced the detection sensitivity and subsequently the LOD of the desired analyte. The continuous exploration of novel nanomaterials and nanostructures will open more windows in pursuit of a highly sensitive SPR sensor. Sensing performance can be further improved with the use of microfluidic chips for efficient liquid manipulation and control of small amounts of the sample, resulting in an automated high-throughput system. Last but not least, there is a new development trend and thus the large demand and opportunity for miniaturized, portable and affordable SPRi instrument in the applications of the POC use. The practical SPRi POC biosensors will be beneficial to end-users, especially medical experts for rapid disease screening in routine biomedical testing, and environmental workers for on-site real-time monitoring of hazardous gas and food contamination. The current SPRi technique still faces some challenges and limitations, for instance, in terms of inefficient sample manipulation method, complicated surface functionalization process and signal contamination due to non-specific binding of molecules. We believe that in the near future, these obstacles will be overcome by a continuously growing research interest in the SPR field.

## Figures and Tables

**Figure 1 sensors-19-01266-f001:**
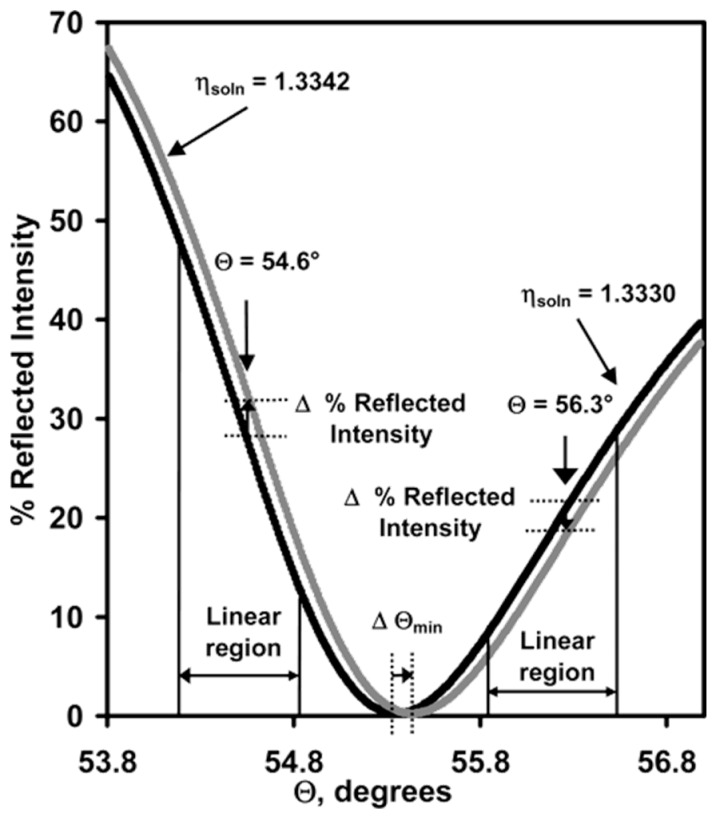
Illustration of the working principle of reflectivity-based surface plasmon resonance imaging (SPRi). Shown here is SPR reflectivity curves versus the incident angle for two samples with different refractive indices in contact with the gold surface. Reprinted with permission from [26].

**Figure 2 sensors-19-01266-f002:**
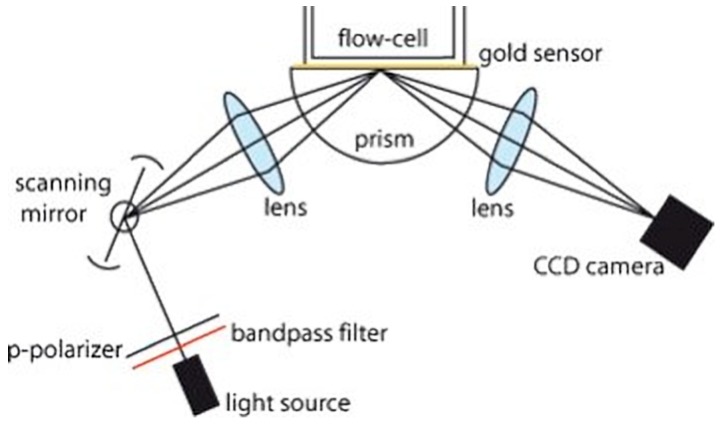
Optical setup of the angle-resolved SPRi. Reprinted with permission from [35].

**Figure 3 sensors-19-01266-f003:**
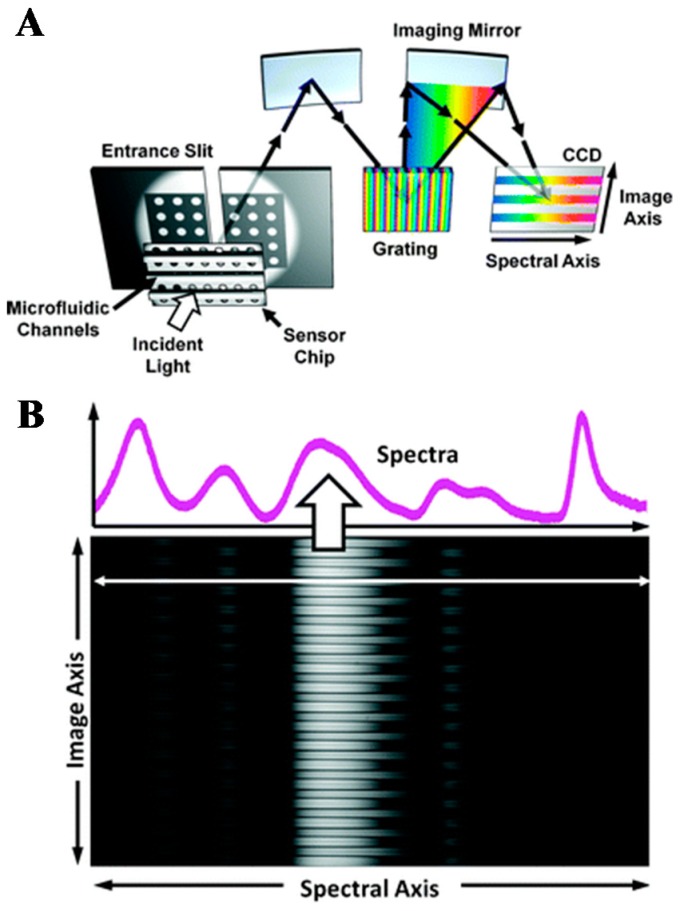
(**A**) Spectral SPRi setup. (**B**) A sample image recorded on the CCD camera. A single horizontal cross-section line shows the transmission spectrum from a single channel. Reprinted with permission from [45].

**Figure 4 sensors-19-01266-f004:**
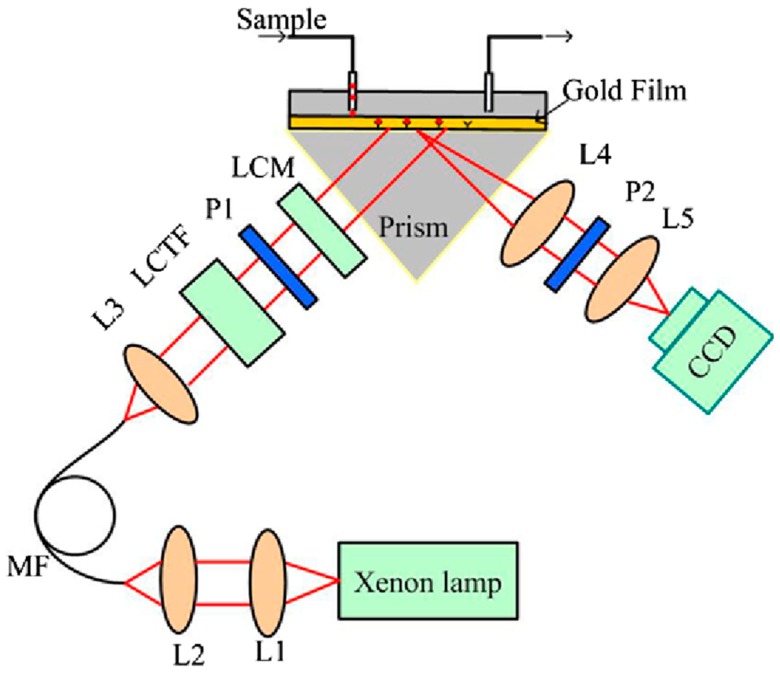
Schematic of wavelength-multiplexing phase-sensitive SPRi system. Reprinted with permission from [53].

**Figure 5 sensors-19-01266-f005:**
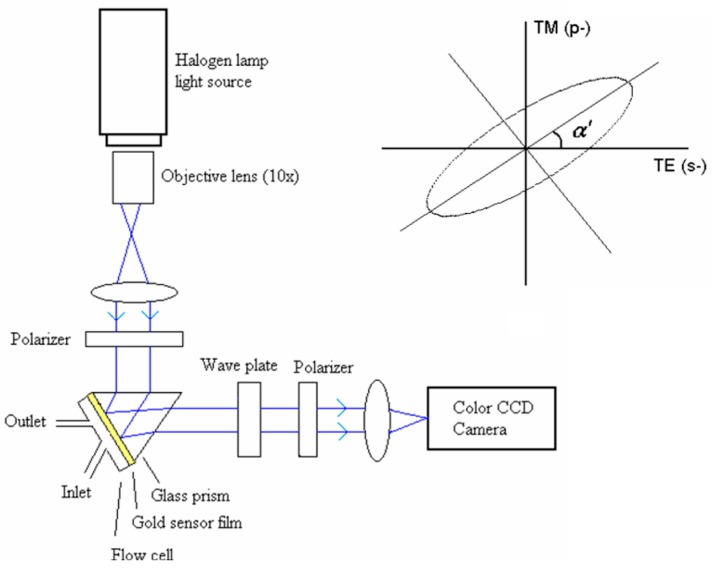
Experimental scheme of the spectral-phase SPRi sensor based on the polarization control scheme. Inset: The orientation angle shift α′ of the ellipse of the light, which is produced by the phase difference between the p-polarization and s-polarization occurring at surface plasmon excitation. Reprinted with permission from [61].

**Figure 6 sensors-19-01266-f006:**
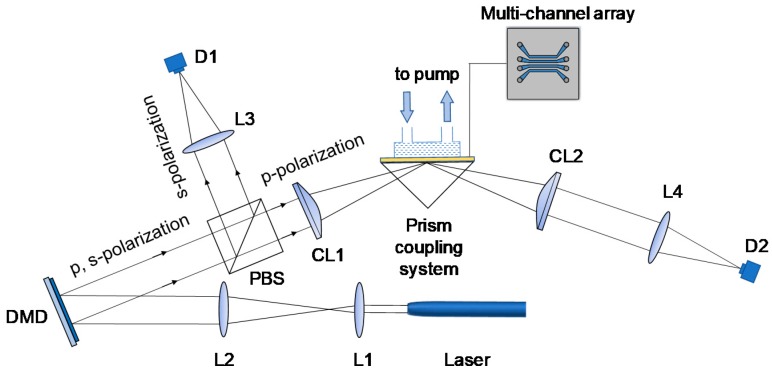
Optical configuration of the DMD-enabled SPRi system based on angular interrogation. Reprinted with permission from [74].

**Figure 7 sensors-19-01266-f007:**
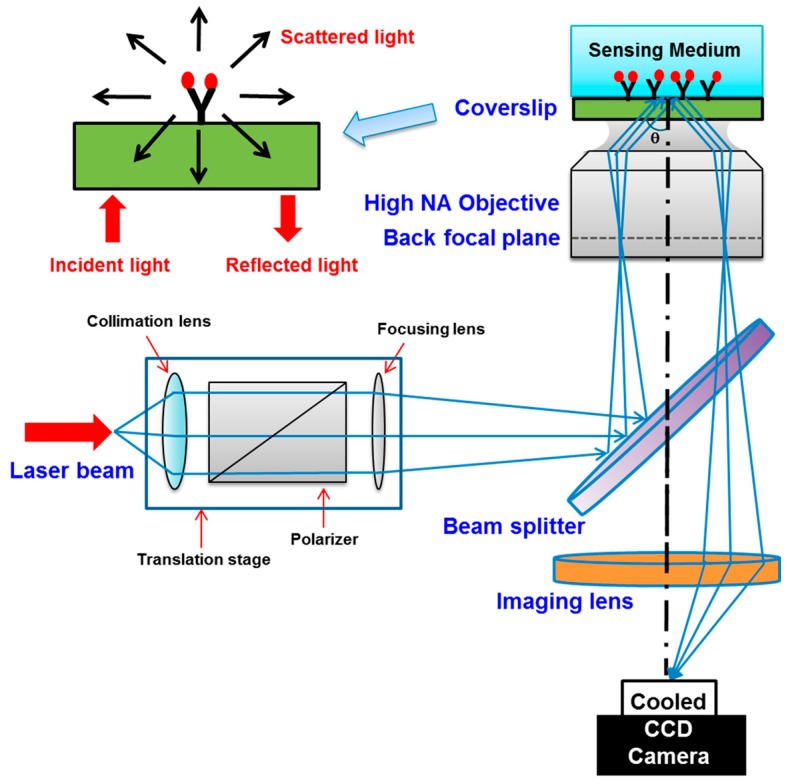
The optical configuration of the objective-type SPRi microscope. Reprinted from [78].

**Figure 8 sensors-19-01266-f008:**
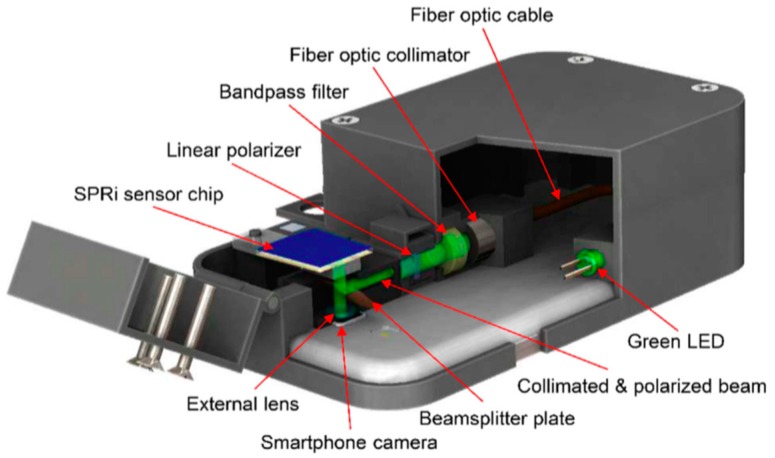
Illustration of SPRi platform integrated with a smartphone. Reprinted with permission from [124].

**Figure 9 sensors-19-01266-f009:**
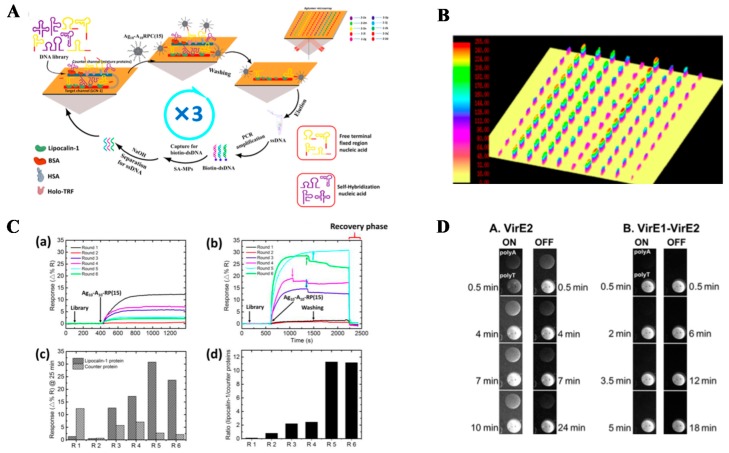
(**A**) Schematic diagram of SPRi-SELEX for anti-LCN-1 aptamers screening. In the SELEX cycle, the nucleic acid library is introduced for positive screening, followed by the introduction of Ag10-A10-RP(15), and the hybridization reaction with a nucleic acid absent of self-hybridization at the tail end. The nucleic acids that have a weak binding affinity with Ag10-A10-RP(15) and some nucleic acids with a weak affinity are washed away. The aptamer candidates are eluted, and PCR amplification is performed. (**B**) SPRi measurements of LCN-1 kinetics onto a ten-component aptamer microarray. The 3D-rendered surface plot of the array showed the difference in SPRi signals after the addition of the LCN-1. (**C**) Evaluation of the nucleic acid- LCN-1 interaction in each round of pools. (a) and (b) Sensorgram for negative and positive selection from R1 to R6 for injection of the pool in each round; (c) and (d) Bar plot of the reflectivity change and ratio of LCN-1 targets and counter proteins on selection in 6 rounds. (**D**) SPRi images of binding and dissociation on (A) VirE2 and (B) VirE1– VirE2. Time duration show the images from the start of the protein (ON) and from the protein-free buffer (OFF) injection. Reprinted with permission from [132,134].

**Figure 10 sensors-19-01266-f010:**
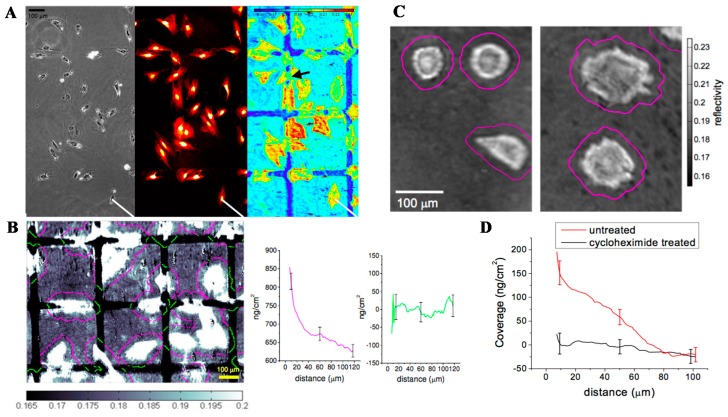
(**A**) Phase contrast (left frame), Texas Red fluorescence staining (middle frame) and SPR imaging (right frame) of fixed vascular smooth muscle (vSMC) after plating on fibronectin separated by 50 μm lines of PEG-thiol, with a color scale bar showing the reflectivity values. The SPRi displays distinct intensity differences between PEG-thiol regions (dark blue), areas of fibronectin (light blue) and cell-substrate contacts (green to red). (**B**) Quantitation of protein deposition at the cell periphery after 24 h in culture using SPRi. Image analysis is used to quantify the protein deposition at the cell periphery after 24 h in culture by dilating the cell contours to extract protein mass versus distance from the cell edge in a region where 18% of the area is occupied by the cells (approximately 25 cells per field of view). The magenta-colored contours overlay fibronectin regions; green contours overlay PEG-thiol regions. The lines shown around the cells correspond to a distance of 40 μm from the cell edges. The protein coverage in ng/cm^2^ versus distance from the cell edge versus distance from the cell edge (magenta) and PEG-thiol regions (green) are determined from the average reflectivity values for each sequential contour. The error bars represent the standard deviation from 4 different sample regions. (**C**) Image analysis comparison of cell deposited materials on the cell periphery for cycloheximide-treated cells compared to untreated cells. Representative SPR images show the vSMCs after plating and with (left) and without (right) exposure to cycloheximide as comparison. Each image is overlaid with contour outlines depicting the distance-to-cell-edge coverage analysis routine from the cell edge. (**D**) The analyzed coverage versus distance-to-cell-edge data on vSMC with and without cycloheximide treatment. The data represent the average of 5 different 500 μm × 500 μm fibronectin patterned regions and the error bars represent the standard deviation. The spatial scale bar indicating 100 μm applies to all the images. Reprinted from [143].

**Figure 11 sensors-19-01266-f011:**
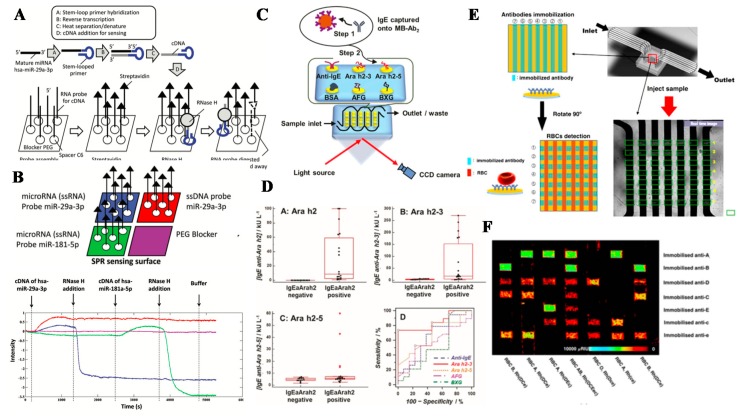
(**A**) The schematic design and workflow for microRNA sensing. First, mature microRNAs are converted into cDNAs by stem-loop primers (top A to D). On the gold SPR surface (bottom), biotinylated-microRNA probes (single-stranded RNA molecules) with spacer C6 are immobilized through thio-linkage, with the unbound region saturated with the blocker PEG. Streptavidins are then added to bind to the immobilized microRNA probes. Hybridization of target cDNAs generates RNA–cDNA hybrids and causes an increase in the SPR signal. RNase H digests the RNA probes while the cDNAs remain intact and bind to new probes for more RNase H digestion. The reaction cycle continues and results in a continuous decrease in SPR signals. (**B**) Specificity of the probe on target microRNA-converted cDNA detection. The probes (ssRNA) hsa-miR-29a-3p (blue), ssDNA probe miR-29a-3p (red), hsa-miR-181-5p (green) or PEG only (purple) immobilized on the SPR sensing surface is subjected to the addition of cDNA of hsa-miR-29a-3p and hsa-miR-181-5p, followed by RNase H loading, at the indicated time points. The signal change compared with the blank control (purple) reveals the specificity for the probe-target binding as well as the RNase H action on RNA-DNA and DNA-DNA hybrids. (**C**) Detection of epitope-specific IgE antibodies, first captured by MB- Ab2. SPRi responses measure individual array spots that have been immobilized with specific allergenic epitopes. (**D**) Distributions of IgE antibody levels in individual levels of IgE anti-Ara h2 (IgEaAra h2) against A) Ara h2, B) Ara h2–3 and C) Ara h2–5; D) ROC curves for peanut sensitization (**E**) The illustration and image of crossing microfluidic microarray fabrication. (**F**) SPRi of the antibody array in each row immobilized with seven antibodies (1–7, horizontal), with each channel showing different RBC blood samples (1–7, vertical). Reprinted with permission from [149,150,153].

**Figure 12 sensors-19-01266-f012:**
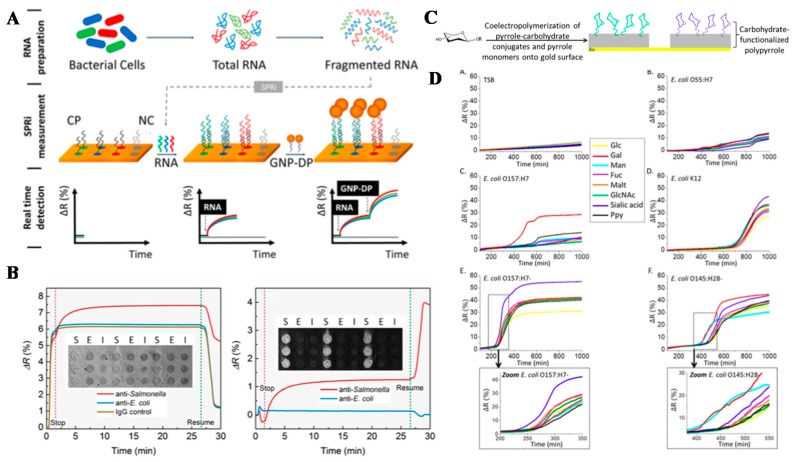
(**A**) Illustration of the assay on sample processing and sensing three specific bacterial strains: L. pneumophila (LP), S. typhimurium (ST), and P. aeruginosa (PA). (**B**) SPRi images (inset), raw (left) and processed (right) sensorgrams of Salmonella detection in chicken rinse matrix in the SPR antibody array (S: anti-Salmonella spots; E: anti-E. coli spots; I: IgG control spots). (**C**) Structures of different pyrrole−carbohydrates conjugates and their fabrication into carbohydrate microarrays. (**D**) SPRi signals of the carbohydrate microarray on screening (A) the buffer (control), (B) E. coli O55:H7, (C) E. coli O157:H7, (D) E. coli K12, (E) E. coli O157:H7^−^, and (F) E. coli O145:H28^−^. Reprinted with permission from [86,159,160].

**Figure 13 sensors-19-01266-f013:**
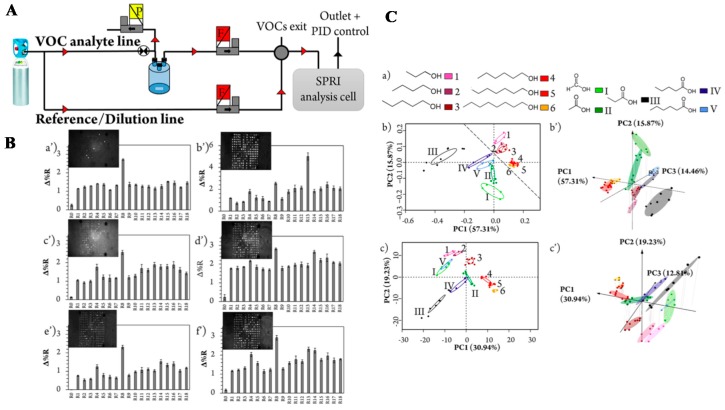
(**A**) Schematic illustration of the SPRi setup, which involves a fluid bench for gas generation, sampling and regeneration, of the optoelectronic nose. (**B**) Characteristic SPRI images of response patterns at equilibrium for (a′) 2-methypyrazine; (b′) phenol; (c′) isoamyl butyrate; (d′) 1-pentanoic acid; (e′) 1-pentanol; and (f′) 1-octanol. (**C**) PCA score plots for the discrimination of structurally similar VOCs. (a) alcohol (1–6) and carboxylic acid family (I–V) with various carbon chain lengths, labeled with different color tints. (b) and (b′) PCA performed with the equilibrium database by (b) using the two and three first principal components respectively. (c) and (c′) PCA conducted based on the kinetic database by using the two and three first principal components respectively. Percentages of the variability associated with each principal component are shown along each axis. Reprinted with permission from [162].

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
