# Peer review of "Recent Advances in Surface Plasmon Resonance Imaging Sensors"

_sensors, 2019, doi:10.3390/s19061266_

Round 1

Reviewer 1 Report

The paper Advances in Surface Plasmon Resonance Imaging Sensors of Wang D. et al.  is a comprehensive review of the field of SPRi. It contributes to the field but in my opinion the balance is not optimal. Let me explain: paragraph 1.2 Operation Principle of SPR Biosensing line 45 is from general SPR theory and can be found various books including the new 2nd edition of the "Handbook of Surface Plasmon Resonance" RSC London 2017. Better is to refer to this book and skip a lot of text. Except: For SPRimaging the propagation length of the plasmon wave and optical lateral resolution are extremely important to sense small objectives cells etc. Interesting discrepancies are that short wavelength generated surface plasmons show optimal lateral resolution but worse sensitivity with wide SPR dips. In contrary, long wavelength SPR show sharp SPR dips but the lateral resolution results in blurry images. Please focus to the theoretical aspect of the lateral resolution of SPR imaging which is the key difference of SPR-imaging with respect to SPR.

Next paragraph 2. Optical configurations... is important for the review and I like the many approaches that are described!

Typo:  Line 141:  "Shaper SPR ...." should be "Sharper SPR...."

Also paragraph 3: "Recent developments...." contribute to the field of SPRimaging including the LSPR instrumentation using nanoparticles and the smartphone-based SPRi.

Cornerstones of SPR are optics,fluidics and surface chemistries. The report neglects to describe the fluidics of SPRimaging. It is not always a simple flow cell with continuous flow but controlled injection, stop flow to allow sedimentation, sometimes back and forth flow etc. Fluidics and surface chemistries are definitely important for SPRi sensing.This is another unbalance in the review paper. 

Regarding par 4. Biosensing applications: Perhaps the authors were not aware of the recent review paper "Trends in SPR Cytometry: Advances in Label-Free Detection of Cell Parameters of Schasfoort et al. Here SPRi is the key technology and cell surface antigens are measured using a so-called T/S detection strategy with flow and stopped-flow including the affinity measurement of cell surface antigens using a gradual increase of the flow velocity. Please add the reference for a better coverage of the review paper. Your reference is 141 but Figure 10E and F is not from Szittner et al. Please correct and describe the T/S detection method. 

In line 480 the authors describe the importance of cell-cell interactions. (I do not agree with cell-cell. but replace by cell-surface interactions).  SPRi is more suitable for cell surface antigen interactions because once a cell (including bacteria) is immobilized it covers the evanescent field completely and another cell binding on top of the first cell layer cannot be detected anymore. Again ref 141 is important...

In line 499 Please change the numbers into 601 +/- 72 g/mL and 140 +/- 25 g/mL

4.2 Healthcare testing: I like the method using RNAseH. It is an elegant way to observe indirect the presence of specific RNA binding. Please start a new alinea at line 544 and 547. Also enter at line 550

Again..... Line 591: Figure 10E and F does not fit with ref 141. 

I don't have specific comments on environmental and gas sensing. 

In the conclusion: The authors are optimistic about POC testing, but in the review there is  not any application where kinetic data measured with SPRi makes a difference with respect to other competing technologies. However, I agree with the authors that SPRimaging is an important upcoming field and definitely one can expect new interesting devices, methods and applications.

I encourage the authors to correct at least the comments above.  Thank you.

Author Response

Point-by-point response to reviewers’ comments:

We wish to express our sincerely gratitude to the valuable comments and suggestions from the reviewers.

Reviewer #1:

1. The paper Advances in Surface Plasmon Resonance Imaging Sensors of Wang D. et al. is a comprehensive review of the field of SPRi. It contributes to the field but in my opinion the balance is not optimal. Let me explain: paragraph 1.2 Operation Principle of SPR Biosensing line 45 is from general SPR theory and can be found various books including the new 2nd edition of the "Handbook of Surface Plasmon Resonance" RSC London 2017. Better is to refer to this book and skip a lot of text. Except: For SPRimaging the propagation length of the plasmon wave and optical lateral resolution are extremely important to sense small objectives cells etc. Interesting discrepancies are that short wavelength generated surface plasmons show optimal lateral resolution but worse sensitivity with wide SPR dips. In contrary, long wavelength SPR show sharp SPR dips but the lateral resolution results in blurry images. Please focus to the theoretical aspect of the lateral resolution of SPR imaging which is the key difference of SPR-imaging with respect to SPR.

Response:

We have cited this book and shortened the theoretical explanation of SPR principle in section 1.2. We also added a paragraph which focuses on the lateral resolution of SPRi.

2. Next paragraph 2. Optical configurations... is important for the review and I like the many approaches that are described! Typo: Line 141: "Shaper SPR ...." should be "Sharper SPR....". Also paragraph 3: "Recent developments...." contribute to the field of SPRimaging including the LSPR instrumentation using nanoparticles and the smartphone-based SPRi.

Response:

Thanks for the comments. We have corrected the typo in the manuscript.

3. Cornerstones of SPR are optics, fluidics and surface chemistries. The report neglects to describe the fluidics of SPRimaging. It is not always a simple flow cell with continuous flow but controlled injection, stop flow to allow sedimentation, sometimes back and forth flow etc. Fluidics and surface chemistries are definitely important for SPRi sensing. This is another unbalance in the review paper.

Response:

We have added two paragraphs emphasizing the importance of fluidics for SPRi in section 3.3 and 3.4 respectively.

4. Regarding par 4. Biosensing applications: Perhaps the authors were not aware of the recent review paper "Trends in SPR Cytometry: Advances in Label-Free Detection of Cell Parameters of Schasfoort et al. Here SPRi is the key technology and cell surface antigens are measured using a so-called T/S detection strategy with flow and stopped-flow including the affinity measurement of cell surface antigens using a gradual increase of the flow velocity. Please add the reference for a better coverage of the review paper. Your reference is 141 but Figure 10E and F is not from Szittner et al. Please correct and describe the T/S detection method.

Response:

We have added the reference and described the T/S detection method. We have also revised the references and the figure numbers in the manuscript.

5. In line 480 the authors describe the importance of cell-cell interactions. (I do not agree with cell-cell. but replace by cell-surface interactions). SPRi is more suitable for cell surface antigen interactions because once a cell (including bacteria) is immobilized it covers the evanescent field completely and another cell binding on top of the first cell layer cannot be detected anymore. Again ref 141 is important...

Response:

We have revised the word "cell-cell" to "cell-surface" interaction.

6. In line 499 Please change the numbers into 601 +/- 72 g/mL and 140 +/- 25 g/mL.

Response:

We have changed the numbers accordingly.

7. 4.2 Healthcare testing: I like the method using RNAseH. It is an elegant way to observe indirect the presence of specific RNA binding. Please start a new alinea at line 544 and 547. Also enter at line 550.

Response:

We have revised the paragraphing.

8. Again..... Line 591: Figure 10E and F does not fit with ref 141.

Response:

We have revised the figure number as 11E and F, which fits with ref 141.

9. I don't have specific comments on environmental and gas sensing. In the conclusion: The authors are optimistic about POC testing, but in the review there is not any application where kinetic data measured with SPRi makes a difference with respect to other competing technologies. However, I agree with the authors that SPRimaging is an important upcoming field and definitely one can expect new interesting devices, methods and applications.

Response:

We have revised the conclusion to focus on the POC testing beneficial to the medical experts and environmental workers on rapid on-site screening, rather than basic research on kinetic study in molecular sensing.

I encourage the authors to correct at least the comments above.  Thank you.

Reviewer #2:

The manuscript by Wang et al. provides a state-of-the-art review of the development in Surface Plasmon Resonance Imaging sensors. The article is well-written and offers a good overview of the last progress in this area, accompanied by a useful description of the SPRi principle and the different operative platforms that have been developed so far. Recent advances are described from the two main points of view: the incorporation of modern instrumentation and technologies, and the emerging applications in different fields of biosciences. Overall, the review is acceptable for publishing in Sensors. I have minor suggestions to the authors:

1. Section 3.3 refers to the incorporation of nanotechnologies to SPRi, describing two main strategies that enhance the sensing performances: the use of nanoparticles for signal amplification, and the use of nanostructured substrates. The latter is only mentioned briefly and as a future perspective for SPRi. However, the use of nanoplasmonics for imaging has been extensively studied nowadays and a lot of applications have been demonstrated; together with important advantages compared to conventional SPRi, such as the effective light coupling without the need of prisms or the easy adaptation to multiplexed assays. It would be good to extend the discussion in this regard and specify how nanoplasmonics is already improving the development and implementation of SPRi.

Response:

We have extended the discussion with the emphasis on the merit of highly miniaturization and capability of multiplexed assay of nanoplasmonics for SPRi in section 3.3.

2.  Section 4.1 is a bit too long and includes very different topics, not all of them relevant or fitting the title “Molecular Sensing”. For example, from line 429 to 439, the authors mention experiment with no especial relevance in biosciences (BSA-antiBSA, or IgG-antiIgG assays), which are usually performed as proof-of-concept assays. Instead, it would be good if the authors search and comment on articles describing applications such as affinity analysis in drug testing or biomarker discovery, nucleic acid analysis in genomics or epigenomics, etc. The application of SPRi for aptamer evaluation is a good example of molecular sensing, but it is not necessary to provide such an explanation about aptamers. At the end, the authors discuss the utility of SPRi for cell analysis (cell-cell interactions, cell activity, etc.). It would be good if this part is separated as a different section and not included in Molecular Sensing.

Response:

We have revised the content to address more studies in drug screening and nucleic acid analysis. Also, we have revised the explanation about the aptamer to be shorter without affecting its introductory understanding. We have separated the SPRi for cell analysis into section "3.2. Live Cell Analysis".

3. Finally, in Conclusions section, I miss a critical discussion that also includes the limitations of SPRi, for example in terms of sample manipulation, surface functionalization, etc.

Response:

We have included the current limitations and challenges of SPRi in the conclusion section. The added part reads: “The current SPRi technique still faces some challenges and limitations, for instance, in terms of inefficient sample manipulation method, complicated surface functionalization process and signal contamination due to non-specific binding of molecules. We believe that in near future these obstacles will be overcome by continuously growing research interests in SPR field.”

Reviewer 2 Report

The manuscript by Wang et al. provides a state-of-the-art review of the development in Surface Plasmon Resonance Imaging sensors. The article is well-written and offers a good overview of the last progress in this area, accompanied by a useful description of the SPRi principle and the different operative platforms that have been developed so far. Recent advances are described from the two main points of view: the incorporation of modern instrumentation and technologies, and the emerging applications in different fields of biosciences. Overall, the review is acceptable for publishing in Sensors. I have minor suggestions to the authors:

-          Section 3.3 refers to the incorporation of nanotechnologies to SPRi, describing two main strategies that enhance the sensing performances: the use of nanoparticles for signal amplification, and the use of nanostructured substrates. The latter is only mentioned briefly and as a future perspective for SPRi. However, the use of nanoplasmonics for imaging has been extensively studied nowadays and a lot of applications have been demonstrated; together with important advantages compared to conventional SPRi, such as the effective light coupling without the need of prisms or the easy adaptation to multiplexed assays. It would be good to extend the discussion in this regard and specify how nanoplasmonics is already improving the development and implementation of SPRi.

-          Section 4.1 is a bit too long and includes very different topics, not all of them relevant or fitting the title “Molecular Sensing”. For example, from line 429 to 439, the authors mention experiment with no especial relevance in biosciences (BSA-antiBSA, or IgG-antiIgG assays), which are usually performed as proof-of-concept assays. Instead, it would be good if the authors search and comment on articles describing applications such as affinity analysis in drug testing or biomarker discovery, nucleic acid analysis in genomics or epigenomics, etc. The application of SPRi for aptamer evaluation is a good example of molecular sensing, but it is not necessary to provide such an explanation about aptamers. At the end, the authors discuss the utility of SPRi for cell analysis (cell-cell interactions, cell activity, etc.). It would be good if this part is separated as a different section and not included in Molecular Sensing.

-          Finally, in Conclusions section, I miss a critical discussion that also includes the limitations of SPRi, for example in terms of sample manipulation, surface functionalization, etc.

Author Response

(The authors gave the same response as above.)

Round 2

Reviewer 1 Report

The authors adequately corrected the comments and I am satisfied with the overall result.

In my opinion the manuscript is almost ready for publication. Some spell check error corrections are required for the minor revision. E.g. the revised final sentence has 4 article errors. "" We believe that in THE near future these obstacles will be overcome by A continuously growing research interest(S) in THE SPR field.  """" .

Author Response

Point-by-point response to reviewer’ comments:

We wish to express our sincerely gratitude to the valuable comments and suggestions from the reviewer.

Reviewer #1:

1. The authors adequately corrected the comments and I am satisfied with the overall result.

In my opinion the manuscript is almost ready for publication. Some spell check error corrections are required for the minor revision. E.g. the revised final sentence has 4 article errors. "" We believe that in THE near future these obstacles will be overcome by A continuously growing research interest(S) in THE SPR field.

Response:

Thanks for the comments. We have corrected the errors in the revised manuscript.